# Learning Efficient Image Super-Resolution Networks via Structure-Regularized Pruning

**Yulun Zhang**[1,†]    **Huan Wang**[1,†,*]    **Can Qin**[1]    **Yun Fu**[1]
[1]Northeastern University, USA

## Abstract

Several image super-resolution (SR) networks have been proposed for efficient SR, achieving promising results. However, they are still not lightweight enough and neglect to be extended to larger networks. At the same time, model compression methods (i.e., knowledge distillation and neural architecture search) usually need heavy extra resources. Then, we turn to a cheap and effective model compression way: network pruning. However, it is not easy to directly apply network pruning to image SR. It is well-known tricky to conduct filter pruning for residual blocks commonly used in SR networks. To solve these problems, we propose structure-regularized pruning (SRP), which imposes regularization on the pruned structure to ensure the locations of pruned filters are aligned across different layers. Specifically, for the layers connected by the same residual, we select the filters of the same indices as unimportant filters. To transfer the expressive power in the unimportant filters to the rest of the network, we employ $L_2$ regularization to drive the weights towards zero so that eventually, their absence will cause minimal performance degradation. We apply SRP to train efficient image SR networks, resulting in a lightweight network SRPN-Lite and a very deep one SRPN. We provide extensive comparisons with both lightweight and larger image SR networks. Our SRPN-Lite and SRPN perform favorably against other recent works.

## 1 Introduction

Given a low-resolution (LR) image, single image super-resolution (SR) aims to reconstruct a high-resolution (HR) output. Essentially, as a many-to-one mapping problemimage SR is ill-posed. To tackle this problem, plenty of deep convolutional neural networks (CNNs) (Dong et al., 2014; 2016; Kim et al., 2016b; Zhang et al., 2018c; 2020; 2021) have been investigated to learn the accurate mapping from LR input to the corresponding HR target.

Deep CNN for image SR is first investigated in SRCNN (Dong et al., 2014) and has continuously shown promising SR performance. SRCNN consists of there convolutional (Conv) layers, constraining its expressivity. Kim *et al.* adopted residual learning to increase network depth in VDSR (Kim et al., 2016a) and obtained notable improvements over SRCNN. Lim *et al.* (Lim et al., 2017) simplified residual blocks and built a much deeper network EDSR. Zhang *et al.* (Zhang et al., 2018b) proposed a even much deeper one RCAN with the residual in residual structure. Empowered by increased network size, deep SR models like EDSR (Lim et al., 2017) and RCAN (Zhang et al., 2018b) have seen remarkable SR performance. However, as a cost, the large model size brings about problems such as excessive memory footprint, slow inference speed. It is thereby impractical to deploy them on resource-constrained platforms directly (Lee et al., 2020).

Aiming for efficient SR, more and more works introduce lightweight network architectures (Ahn et al., 2018; Luo et al., 2020). Ahn *et al.* proposed cascading residual network (CARN) (Ahn et al., 2018). Hui *et al.* proposed information multi-distillation network (IMDN) (Hui et al., 2019). Lee *et al.* introduced knowledge distillation (Hinton et al., 2014) for image SR (Lee et al., 2020). Besides, neural architecture search (NAS) (Zoph & Le, 2017; Elsken et al., 2019) was also utilized for lightweight SR model (Chu et al., 2019a). However, there are still several downsides to these networks: **(1)** There is still an obvious performance gap between those lightweight models and the very deep models; **(2)** These methods can consume considerable computation resources for training. For example, (Chu et al., 2019a) train a single network with 8 Tesla V100 GPUs, and the training

---

[†]Equal contribution. The main work was done when Yulun Zhang was at Northeastern University.
[*]Corresponding author: Huan Wang (wang.huan@northeastern.edu)

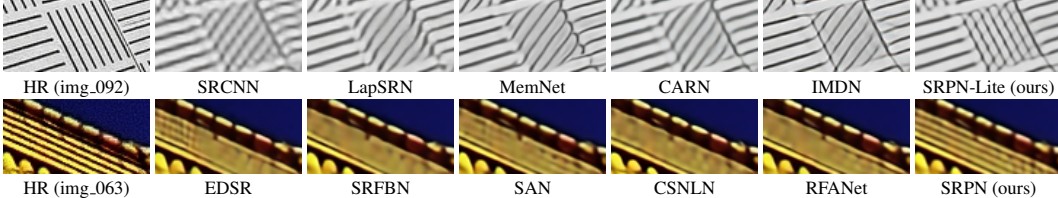

Figure 1: Visual comparisons (×4) about lightweight and large SR networks on Urban100 dataset. The first row is lightweight SR networks comparison. The second row is about larger models comparison. SRPN-Lite and SRPN are our proposed lightweight and larger image SR networks.

takes about three days; **(3)** It is hard for most lightweight SR network designs to generalize to more large-scale networks and achieve superior performance at the same time. So, it is needed to explore more resource-friendly, effective, and generic lightweight SR networks.

On the other hand, neural network pruning is well-known as an effective technique to reduce model complexity (Reed, 1993; Sze et al., 2017). For acceleration, filter pruning (a.k.a. structured pruning) (Li et al., 2017) attracts more attention than weight-element pruning (a.k.a. unstructured pruning) (Han et al., 2015; 2016b). Introducing filter pruning into image SR is a promising way to achieve a good trade-off between performance and complexity. However, it is not easy to apply filter pruning techniques to image SR networks directly. This is mainly because residual connections are well-known difficult to prune in structured pruning (Li et al., 2017). On the other hand, they are *extensively* used in state-of-the-art (SOTA) image SR methods (e.g., EDSR (Lim et al., 2017) has 32 residual blocks; RCAN (Zhang et al., 2018b) even has nested residual blocks).

To tackle the above issue, we propose *Structure-Regularized Pruning* (SRP), which imposes regularization on the pruned structure to ensure the locations of pruned filters are *aligned* across different layers. Specifically, for the layers connected by the same residual, we select the filters of the same indices as unimportant filters (i.e., those we will remove finally). To transfer the expressive power in the unimportant filters to the remainder of the network, we employ $L_2$ regularization to drive the weights towards zero gradually so that eventually, their absence will incur negligible performance loss. To the best of our knowledge, our SRP is one of the leading works (Zhang et al., 2021) to leverage structured pruning for efficient image SR. Our main contributions are as follows:

- We propose a network structure-regularized pruning (SRP) method to learn efficient image SR networks. We try to jointly train image SR models with network pruning simultaneously to achieve high reconstruction performance as well as efficiency.

- Our SRP provides a general idea to structurally prune networks, which consists of extensive residual connections. The introduction of regularization as a pruning tool manages to maintain the expressivity of the original network while peeling off the unnecessary redundancy.

- We employ SRP to train efficient image SR networks, resulting in a lightweight network (named SRPN-Lite) and a very deep one (named SRPN). We achieve superior performance gains on both lightweight and large image SR networks.

## 2 RELATED WORK

**Deep Image SR Models.** Dong *et al*. (Dong et al., 2014) firstly proposed SRCNN for image SR and achieved superior performance with only 3 convolutional (Conv) layers. Residual learning was introduced in VDSR (Kim et al., 2016a), reaching 20 Conv layers and a significant improvement over SRCNN. Tai *et al*. later introduced memory block in MemNet (Tai et al., 2017b) for deeper network structure. Lim *et al*. (Lim et al., 2017) simplified the residual block (He et al., 2016) and constructed deeper and wider networks with a large number of parameters. Zhang *et al*. (Zhang et al., 2018b) proposed an even deeper one, residual channel attention network (RCAN), where the attention mechanism was firstly introduced in image SR. Liu *et al*. proposed FRANet (Liu et al., 2020) to make the residual features focus on critical spatial contents. Later, Zhang *et al*. (Zhang et al., 2019) extended to image restoration with residual non-local attention. Mei *et al*. proposed CSNLN (Mei et al., 2020) by combining local and non-local feature correlations. Most of those methods have achieved SOTA results. However, they suffer from huge model size (i.e., network parameter number) and/or heavy computation operations (i.e., FLOPs).

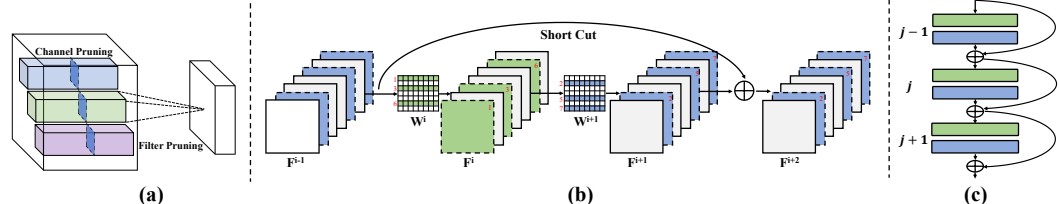

Figure 2: **(a)** Illustration of channel-wise and filter-wise pruning for single Conv layer. In this work, we adopt filter-wise pruning to learn efficient image SR networks. **(b)** Illustration of filter pruning *within* a residual block. We depict deep features $F$ as 3d cubes. We expend the Conv kernel $W$ (4d tensor) as a 2d matrix here for easy illustration (each row represents a filter). Both green and blue color denote the *pruned* filters (or feature map channels): green denotes the pruned filters in *free Conv layers*; blue denotes the pruned filters in *constrained Conv layers*. The basic idea of our SRP is to apply the $L_2$ regularization to the unimportant filters to make sure the pruned indices of $F^{(i)}$ are exactly the same as those of $F^{(i+2)}$. **(c)** Illustration of filter pruning across different residual blocks. All the layers that are directly followed by the add operators are constrained Conv layers (in blue). Others are free Conv layers (in green).

**Lightweight Image SR Models.** Recent years have been rising interest in investigating lightweight image SR models. These approaches try to design lightweight architectures, which mainly take advantage of recursive learning and channel splitting. Kim *et al.* firstly decreased parameter number by utilizing recursive learning in DRCN (Kim et al., 2016b). Ahn *et al.* proposed CARN by designing a cascading mechanism upon a residual network (Ahn et al., 2018). Hui *et al.* proposed IMDN by using distillation and fusion modules (Hui et al., 2019). Luo *et al.* designed the lattice block with butterfly structures (Luo et al., 2020). Recently, neural architecture search was applied for image SR, like FALSR (Chu et al., 2019a). Also, model compression techniques have been explored for image SR. He *et al.* proposed knowledge distillation based feature-affinity for efficient image SR (He et al., 2020). Lee *et al.* distilled knowledge from a larger teacher network to a student one (Lee et al., 2020). Those lightweight image SR models have obtained great progress, but we still need to investigate deeper for more efficient image SR models.

**Neural Network Pruning.** Network pruning aims to eliminate redundant parameters in a neural network without compromising its performance seriously (Reed, 1993; Sze et al., 2017). The methodology of pruning mainly falls into two groups: filter pruning (or more generally known as structured pruning)* and weight-element pruning (also referred to as unstructured pruning). The former aims to remove weights by filters (i.e., 4-d tensors), while the latter removes weights by single elements (i.e., scalars). Structured pruning results in regular sparsity after pruning. It does *not* demand any special hardware features to achieve considerable practical acceleration. In contrast, unstructured pruning leads to irregular sparsity. Leveraging the irregular sparsity for acceleration typically demands special software supports, while past works have shown the practical speedup is very limited (Wen et al., 2016), unless using customized hardware platforms (Han et al., 2016a). In this paper, we tackle *filter pruning* instead of weight-element pruning for effortless acceleration. The major efforts in pruning (mainly in image classification) have been focusing on proposing a more sound pruning criterion to select unimportant weights (Reed, 1993; Sze et al., 2017). Criteria based on weight magnitude (Han et al., 2015; 2016b; Li et al., 2017) are the most prevailing ones, which we will also employ to develop our method in this paper.

## 3 METHODOLOGY

We first show the overview of the problem setting about deep CNN for image SR. It is also observed that excessive redundancy exists in the SR deep CNNs. Then we move on to proposing our *structure-regularized pruning (SRP)* method attempting to achieve more efficient SR networks.

### 3.1 DEEP CNN FOR IMAGE SR

In image super-resolution (SR), a high-resolution (a.k.a. super-resolved) image $I_{SR}$ is reconstructed from its low-resolution (LR) input $I_{LR}$. This image SR process can be described as

$$I_{SR} = \mathcal{F}_{SR}(I_{LR}; \Theta), \tag{1}$$

---

*Filter pruning is a sub-concept within structured pruning, while they are often used interchangeably in the literature. In this paper, we stick to this terminology practice.

where $\mathcal{F}_{SR}(\cdot)$ is the image SR network, parameterized by $\Theta$. Meanwhile, the LR input $I_{LR}$ from the corresponding HR image can be formulated as

$$I_{LR} = \mathcal{F}_{\downarrow_s}(I_{HR}), \tag{2}$$

where $\mathcal{F}_{\downarrow_s}(\cdot)$ downscales the original HR image $I_{HR}$ by a scale factor $s$. This downscaling process typically introduces additional compression, blurring, noise, and/or other unknown artifacts. Therein, the desired structural details could be removed more or less. The job of image SR methods is to reconstruct the high-frequency information as much as possible.

We aim to present a structural sparsity inducing SR network learning method, which can deliver more efficient SR networks with fewer parameters and FLOPs.

### 3.2 STRUCTURE-REGULARIZED PRUNING (SRP)

**Pruned Index Constraint**. Pruning filters in residual networks is well-known non-trivial as the Add operators in residual blocks require the pruned filter indices across different residual blocks must be aligned. A figurative illustration of filter pruning within a residual block is shown in Fig. 2(b). A typical residual block (e.g., in EDSR (Lim et al., 2017), RCAN (Zhang et al., 2018b)) consists of two convolutional layers. According to the mutual connection relationship, the convolutional layers can be categorized into two groups. One group is made up with the layers that can be pruned *without any constraint*, dubbed *free Conv layers* in this work; the other comprises Conv layers whose filters must be pruned *at the same indices*, dubbed *constrained Conv layers*. Concretely, the layer $W^{(i)}$ in Fig. 2(b) is a free Conv layer, while the layer $W^{(i+1)}$ is a constrained one.

Owing to the pruned index constraint problem, many filter pruning methods in image classification simply do *not* prune the last Conv layer in residual blocks (Li et al., 2017; Wang et al., 2021), which can still deliver considerable speedup of practical interest. Nevertheless, this doing-nothing solution can barely translate to the image SR networks if we target a considerable speedup. The root cause is that image SR networks usually utilize *many more* residual blocks, and each block usually has only two Conv layers. In some top-performing SR networks (e.g., RCAN Zhang et al. (2018b)), there are even *nested* residual blocks. Taking EDSR for a concrete example, it has 32 residual blocks with two Conv layers within each block. If the 2nd Conv layer in a residual block is spared from pruning, half of the Conv layers will not be pruned. In other words, *at best*, we can only obtain $2\times$ theoretical acceleration measured by FLOPs reduction.

Given this issue, it is imperative to prune *all* the Conv layers in residual blocks, thus calling for an approach to align the pruned indices of all constrained Conv layers. We then use regularization as a promising solution considering that it has been widely used before to impose priors on the sparsity structure in classification (Reed, 1993; Wen et al., 2016; Wang et al., 2021).

Our SRP method is based on regularization to form desired hardware-friendly sparsity structure. Primarily, there are three questions to answer in SRP: (1) which pruning criterion is used to choose unimportant filters; (2) which kind of regularization is chosen to induce sparsity; (3) how to schedule the pruning process to make the algorithm robust and easy to use.

**(1) Pruning Criterion**. When it comes to formulated pruning criteria, they mainly fall into three groups: magnitude based (Han et al., 2015; Li et al., 2017), 1st-order gradient based (Molchanov et al., 2017; 2019), and 2nd-order gradient based (LeCun et al., 1990; Hassibi & Stork, 1993; Wang et al., 2019a). The 1st-order and 2nd-order gradient-based criteria are based on Taylor approximation of the increased loss when a weight is removed, pioneered by (LeCun et al., 1990). They are more accurate than magnitude-based one right after pruning. However, pruning is typically followed by a finetuning process (Reed, 1993) to regain performance. After finetuning, the advantage of the 1st-order and 2nd-order gradient-based criteria usually becomes marginal. Meanwhile, they are typically more costly than the magnitude-based criteria. Taking the cost and flexibility into account, we simply employ magnitude-based pruning criteria.

Specifically, given a pretrained SR network, for all the free Conv layers, we sort the filters in ascending order by their $L_1$-norms and select the filers with the least norms as *unimportant filters*, by a predefined pruning ratio $r \in (0, 1)$. For the constrained Conv layers, the problem becomes a little bit more complex. Given the pruned index constraint, ideally, we can only prune the filters in the *intersection* set (denoted by $\mathcal{S}^{(net)}$) for the whole network,

$$\mathcal{S}^{(net)} = \bigcap_{l \in \mathcal{C}} S^{(l)}, \tag{3}$$

where $\mathcal{S}^{(l)}$ represents the set of unimportant filters selected by certain pruning criterion in the $l$-th Conv layer; $\mathcal{C}$ denotes the set of constrained Conv layers. For top-performed image SR networks, $|\mathcal{C}|$ ($|\cdot|$ denotes cardinality of a set) is at the order of magnitude of hundreds (e.g., $|\mathcal{C}| = 200$ for RCAN). Meanwhile, given a pretrained model, its pruned filter indices by $L_1$-norm are usually *random* across different layers. As a consequence, $|\mathcal{S}^{(net)}|$ will be rather small (in practice, on RCAN, we even observe $|\mathcal{S}^{(net)}| = 0$ when we try to prune 32 filters out of 64). This issue brings much trouble to user control when we want to prune a layer to a pre-defined number of filters. To address this issue, we choose to *randomly* select a set of filters as unimportant and fixed during the whole pruning process. In practice, we find this simple scheme works pretty well. The reason enabling us to do so is that regularization for pruning is a *gradual* process. It allows ample time for the network to recover from the introduced penalty term (this will be empirically verified in our experiments, see Fig. 3). Apart from aligning cross-layer sparsity structure, this is the second reason we choose regularization for pruning in our algorithm.

**(2) Regularization Form**. Although $L_1$ regularization is well-known to induce sparsity in machine learning, it is hard to tune the coefficient to realize a desired trade-off between sparsity and performance. Instead, $L_2$ regularization is thereby adopted in our method given its more tamable control over the sparsification process – note the gradient of $L_2$ regularization is proportion to the weight magnitude while the gradient of $L_1$ regularization is not. Specifically, given the loss function $\mathcal{L}$ of a neural network $\Theta$, the total error function $\mathcal{E}$ with adaptive $L_2$ regularization term is formulated as

$$\mathcal{E}(\Theta; \mathcal{D}) = \mathcal{L}(\Theta; \mathcal{D}) + \frac{1}{2} \sum_{l,j} \alpha_j^{(l)} ||W_j^{(l)}||_2^2, \tag{4}$$

where $\mathcal{D}$ stands for the training dataset; $W_j^{(l)}$ refers to the $j$-th filter in $l$-th Conv layer; $\alpha_j^{(l)}$ stands for the $L_2$ regularization co-efficient for that filter. If a filter $j$ falls into the set of unimportant filters $S^{(l)}$, $\alpha_j^{(l)} > 0$; otherwise, $\alpha_j^{(l)} = 0$ since we do not restrict the learning of important filters.

**(3) Regularization Schedule**. As mentioned above, for the constrained Conv layers, we select the unimportant filters randomly. To mitigate the side effect of this sub-optimal choice, the network should be provided with abundant training iterations to adapt so that it can transfer its expressive power to the remaining part of the network. To achieve this goal, we choose to gradually increase the penalty strength, drawing inspirations from (Wang et al., 2019b; 2021),

$$\alpha_j^{(l)} = \alpha_j^{(l)} + \Delta, \tag{5}$$

where $\Delta$ (a pre-defined constant) is a small amount that the penalty strength is increased each time. This design allows us to raise the penalty strength to a large amount eventually (like 0.5 in our experiments; in contrast, note the normal $L_2$ regularization co-efficient is at the scale of $10^{-4}$). As a result, the unimportant weights are driven rather close to zero, and removing them will hardly hurt the performance. Besides, $\alpha$ is updated every $T$ iterations during training for stability.

To indicate when to terminate the pruning process, a ceiling limit $\tau$ is introduced for the regularization co-efficient $\alpha$. The pruning is over when all the $\alpha$'s for unimportant filters arise to $\tau$.

### 3.3 LEARNING EFFICIENT IMAGE SR MODELS VIA SRP

SRP can be applied to top-performing SR algorithms in a *plug-and-play* manner – simply replace the loss function $\mathcal{L}$ in Eq. (4) with the loss objective function of an SR approach of interest. The proposed $L_2$ regularization can be implemented on any deep learning framework very easily.

Upon finishing pruning, we take away the unimportant filters (not zeroing out them only, but literally removing them from the model), which will give us a compact SR model. Finally, the compact model will be finetuned to regain performance following the common practice in pruning (Reed, 1993).

### 3.4 IMPLEMENTATION DETAILS

We explain how to apply SRP to both lightweight and deep image SR networks to demonstrate the method can work with a wide range of SR networks under different computation budgets.

**Lightweight Networks.** First, we adapt the EDSR baseline (Lim et al., 2017) with 16 residual blocks and remove its final Conv layer to reduce model size. The reconstruction upscaling is realized by the pixel-shuffle layer (Shi et al., 2016) following common practice. The channel number in the revised EDSR baseline is first set as 256 and then pruned to 45. For $\times 2$ scale, we reduce the number of parameters from 19.5M to 609K and name the compressed model as SRPN-Lite.

| Pruning ratio | Params (K) | FLOPs (G) | Scratch | $L_1$-norm pruning (Li et al., 2017) | SRP (ours) |
|---|---|---|---|---|---|
| 0.1 | 1,101.8 | 254.5 | 37.85 | 37.91 | **37.97** |
| 0.3 | 681.1 | 157.7 | 37.81 | 37.81 | **37.89** |
| 0.5 | 381.8 | 88.9 | 37.75 | 37.73 | **37.84** |
| 0.7 | 154.2 | 36.5 | 37.56 | 37.58 | **37.71** |
| 0.9 | 26.9 | 7.3 | 36.74 | 36.87 | **37.28** |

Table 1: PSNR (dB) values on the Set5 ($\times 2$) between SRP and other two baselines to achieve exactly *the same* compact network. "Scratch" stands for training the small network from scratch. "$L_1$-norm" removes filters with the smallest $L_1$-norms. The unpruned model is EDSR baseline (Params: 1,369.9K, FLOPs: 316.3G, PSNR: 37.99 dB). The best results are **highlighted**.

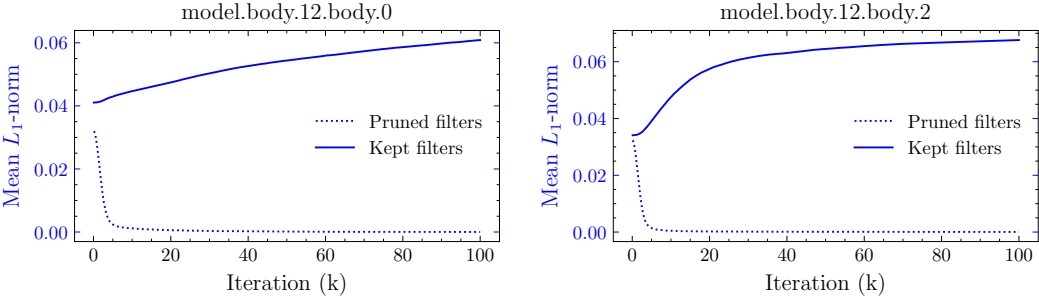

Figure 3: Plots of the mean $L_1$-norm of filters vs. iterations for "model.body.12.body.0" (free Conv layer) and "model.body.12.body.2" (constrained Conv layer) in EDSR baseline. As expected, the unimportant filters ("Pruned filters") are driven down (owing to strong regularization); interestingly, the important ones arise spontaneously.

**Deep Networks.** To apply SRP to the very deep network RCAN (Zhang et al., 2018b), a representative top-performing deep SR network with over 400 Conv layers, we revise RCAN by removing all the channel attention modules (Zhang et al., 2018b). The channel number in the revised RCAN is chosen as 96 and then pruned to 64. For $\times 2$ scale, we reduce the number of parameters from 34.5M to 15.3M and dub the compressed model as SRPN.

## 4 EXPERIMENTAL RESULTS

### 4.1 SETTINGS

**Data and Evaluation.** We use DIV2K dataset (Timofte et al., 2017) and Flickr2K Lim et al. (2017) as training data, following most recent works (Timofte et al., 2017; Lim et al., 2017; Zhang et al., 2018a; Haris et al., 2018). For testing, we use five standard benchmark datasets: Set5 (Bevilacqua et al., 2012), Set14 (Zeyde et al., 2010), B100 (Martin et al., 2001), Urban100 (Huang et al., 2015), and Manga109 (Matsui et al., 2017). The SR results are evaluated with PSNR and SSIM (Wang et al., 2004) on the Y channel in YCbCr space.

**Training Settings.** Following (Zhang et al., 2018b), data augmentation is used in training – training images are randomly rotated by $90°$, $180°$, $270°$ and flipped horizontally. Image patches (patch size $48\times48$) are cropped out to form each training batch. Adam optimizer (Kingma & Ba, 2014) is adopted for training with $\beta_1$=0.9, $\beta_2$=0.999, and $\epsilon$=$10^{-8}$. Initial learning rate is set to $10^{-4}$ and then decayed by factor 0.5 every $2\times10^5$ iterations. We use PyTorch (Paszke et al., 2017) to implement our models with a Tesla V100 GPU[†].

### 4.2 ABLATION STUDY

The EDSR baseline with 16 residual blocks (Lim et al., 2017)[‡] is used as the backbone for ablation study given its broad use in the community.

**Comparison with Baseline Pruning Methods**. To start with, we show that our SRP is more effective than the available baseline compact image SR network learning methods. Two baseline methods

---

[†]The code is available at https://github.com/mingsun-tse/SRP.
[‡]https://github.com/sanghyun-son/EDSR-PyTorch

| Method | Scale | Params | Mult-Adds | Set5 | | Set14 | | B100 | | Urban100 | |
|---|---|---|---|---|---|---|---|---|---|---|---|
| | | | | PSNR | SSIM | PSNR | SSIM | PSNR | SSIM | PSNR | SSIM |
| SRCNN | ×2 | 57K | 52.7G | 36.66 | 0.9542 | 32.42 | 0.9063 | 31.36 | 0.8879 | 29.50 | 0.8946 |
| FSRCNN | ×2 | 12K | 6.0G | 37.00 | 0.9558 | 32.63 | 0.9088 | 31.53 | 0.8920 | 29.88 | 0.9020 |
| VDSR | ×2 | 665K | 612.6G | 37.53 | 0.9587 | 33.03 | 0.9124 | 31.90 | 0.8960 | 30.76 | 0.9140 |
| DRCN | ×2 | 1,774K | 17,974.3G | 37.63 | 0.9588 | 33.04 | 0.9118 | 31.85 | 0.8942 | 30.75 | 0.9133 |
| LapSRN | ×2 | 813K | 29.9G | 37.52 | 0.9590 | 33.08 | 0.9130 | 31.80 | 0.8950 | 30.41 | 0.9100 |
| DRRN | ×2 | 297K | 6,796.9G | 37.74 | 0.9591 | 33.23 | 0.9136 | 32.05 | 0.8973 | 31.23 | 0.9188 |
| MemNet | ×2 | 677K | 2,662.4G | 37.78 | 0.9597 | 33.28 | 0.9142 | 32.08 | 0.8978 | 31.31 | 0.9195 |
| CARN | ×2 | 1,592K | 222.8G | 37.76 | 0.9590 | 33.52 | 0.9166 | 32.09 | 0.8978 | 31.92 | 0.9256 |
| IMDN | ×2 | 694K | 158.8G | 38.00 | 0.9605 | 33.63 | 0.9177 | 32.19 | 0.8996 | 32.17 | 0.9283 |
| SRPN-Lite (ours) | ×2 | 609K | 139.9G | **38.10** | **0.9608** | **33.70** | **0.9189** | **32.25** | **0.9005** | **32.26** | **0.9294** |
| SRCNN | ×3 | 57K | 52.7G | 32.75 | 0.9090 | 29.28 | 0.8209 | 28.41 | 0.7863 | 26.24 | 0.7989 |
| FSRCNN | ×3 | 12K | 5.0G | 33.16 | 0.9140 | 29.43 | 0.8242 | 28.53 | 0.7910 | 26.43 | 0.8080 |
| VDSR | ×3 | 665K | 612.6G | 33.66 | 0.9213 | 29.77 | 0.8314 | 28.82 | 0.7976 | 27.14 | 0.8279 |
| DRCN | ×3 | 1,774K | 17,974.3G | 33.82 | 0.9226 | 29.76 | 0.8311 | 28.80 | 0.7963 | 27.15 | 0.8276 |
| DRRN | ×3 | 297K | 6,796.9G | 34.03 | 0.9244 | 29.96 | 0.8349 | 28.95 | 0.8004 | 27.53 | 0.8378 |
| MemNet | ×3 | 677K | 2,662.4G | 34.09 | 0.9248 | 30.00 | 0.8350 | 28.96 | 0.8001 | 27.56 | 0.8376 |
| CARN | ×3 | 1,592K | 118.8G | 34.29 | 0.9255 | 30.29 | 0.8407 | 29.06 | 0.8034 | 28.06 | 0.8493 |
| IMDN | ×3 | 703K | 71.5G | 34.36 | 0.9270 | 30.32 | 0.8417 | 29.09 | 0.8046 | 28.17 | 0.8519 |
| SRPN-Lite (ours) | ×3 | 615K | 62.7G | **34.47** | **0.9276** | **30.38** | **0.8425** | **29.16** | **0.8061** | **28.22** | **0.8534** |
| SRCNN | ×4 | 57K | 52.7G | 30.48 | 0.8628 | 27.49 | 0.7503 | 26.90 | 0.7101 | 24.52 | 0.7221 |
| FSRCNN | ×4 | 12K | 4.6G | 30.71 | 0.8657 | 27.59 | 0.7535 | 26.98 | 0.7150 | 24.62 | 0.7280 |
| VDSR | ×4 | 665K | 612.6G | 31.35 | 0.8838 | 28.01 | 0.7674 | 27.29 | 0.7251 | 25.18 | 0.7524 |
| DRCN | ×4 | 1,774K | 17,974.3G | 31.53 | 0.8854 | 28.02 | 0.7670 | 27.23 | 0.7233 | 25.14 | 0.7510 |
| LapSRN | ×4 | 813K | 149.4G | 31.54 | 0.8850 | 28.19 | 0.7720 | 27.32 | 0.7280 | 25.21 | 0.7560 |
| DRRN | ×4 | 297K | 6,796.9G | 31.68 | 0.8888 | 28.21 | 0.7720 | 27.38 | 0.7284 | 25.44 | 0.7638 |
| MemNet | ×4 | 677K | 2,662.4G | 31.74 | 0.8893 | 28.26 | 0.7723 | 27.40 | 0.7281 | 25.50 | 0.7630 |
| CARN | ×4 | 1,592K | 90.9G | 32.13 | 0.8937 | 28.60 | 0.7806 | 27.58 | 0.7349 | 26.07 | 0.7837 |
| IMDN | ×4 | 715K | 40.9G | 32.21 | 0.8948 | 28.58 | 0.7811 | 27.56 | 0.7353 | 26.04 | 0.7838 |
| SRPN-Lite (ours) | ×4 | 623K | 35.8G | **32.24** | **0.8958** | **28.69** | **0.7836** | **27.63** | **0.7373** | **26.16** | **0.7875** |

Table 2: Quantitative results of lightweight SR networks. Best results are **highlighted**.

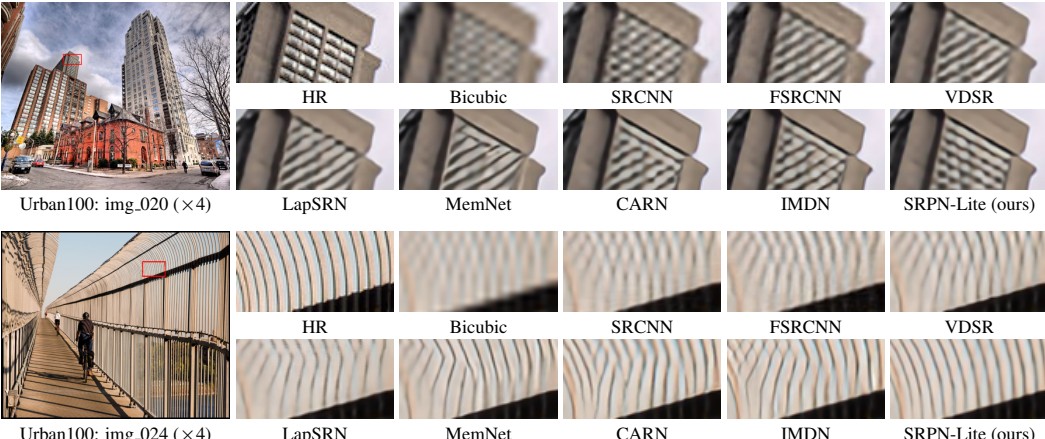

Figure 4: Visual comparison of different *lightweight* SR approaches on the Urban100 dataset (×4).

are compared to: training from scratch and the original $L_1$-norm pruning (Li et al., 2017). The results are presented in Tab. 1. As we can see, our pruned network delivers the best PSNR *consistently* across different pruning ratios. Notably, under a *greater* pruning ratio, the advantage of our method over scratch training or $L_1$-norm pruning is *more evident* in general, suggesting that our SRP is more valuable in aggressive pruning cases. Of particular note is that, our method SRP also utilizes the $L_1$-norm as the scoring criterion to select unimportant filters, same as (Li et al., 2017). Nevertheless, our method delivers significantly better results than theirs. The primary reason is that they do not impose any regularization on the pruned structure; the remaining feature maps are thus *mismatched* in residual blocks after pruning. In contrast, our method SRP is not bothered by this issue, showing the effectiveness of our proposed structure regularization.

**Visualization of Pruning Process**. In Fig. 3, we visualize the pruning process by plotting the mean $L_1$-norm of filters in two layers of EDSR baseline during SRP training. The filters are split into two groups, kept filters and pruned filters. As is shown in the figure, the mean $L_1$-norm of the pruned filters goes down gradually because the penalty grows stronger and stronger, driving them towards zero. Interestingly, note the $L_1$-norms of the kept filters arise themselves. Recall that there is no explicit regularization to promote them to grow. In other words, the network *learns to recover by itself*, akin to the compensation effect found in human brain (Duffau et al., 2003).

| Method | Scale | Set5 PSNR | Set5 SSIM | Set14 PSNR | Set14 SSIM | B100 PSNR | B100 SSIM | Urban100 PSNR | Urban100 SSIM | Manga109 PSNR | Manga109 SSIM |
|---|---|---|---|---|---|---|---|---|---|---|---|
| EDSR | ×2 | 38.11 | 0.9602 | 33.92 | 0.9195 | 32.32 | 0.9013 | 32.93 | 0.9351 | 39.10 | 0.9773 |
| DBPN | ×2 | 38.09 | 0.9600 | 33.85 | 0.9190 | 32.27 | 0.9000 | 32.55 | 0.9324 | 38.89 | 0.9775 |
| RCAN | ×2 | 38.27 | 0.9614 | 34.12 | 0.9216 | 32.41 | 0.9027 | 33.34 | 0.9384 | 39.44 | 0.9786 |
| SRFBN | ×2 | 38.11 | 0.9609 | 33.82 | 0.9196 | 32.29 | 0.9010 | 32.62 | 0.9328 | 39.08 | 0.9779 |
| SAN | ×2 | 38.31 | **0.9620** | 34.07 | 0.9213 | 32.42 | 0.9028 | 33.10 | 0.9370 | 39.32 | 0.9792 |
| HAN | ×2 | 38.27 | 0.9614 | 34.16 | 0.9217 | 32.41 | 0.9027 | 33.35 | 0.9385 | 39.46 | 0.9785 |
| IGNN | ×2 | 38.24 | 0.9613 | 34.07 | 0.9217 | 32.41 | 0.9025 | 33.23 | 0.9383 | 39.35 | 0.9786 |
| CSNLN | ×2 | 38.28 | 0.9616 | 34.12 | 0.9223 | 32.40 | 0.9024 | 33.25 | 0.9386 | 39.37 | 0.9785 |
| RFANet | ×2 | 38.26 | 0.9615 | 34.16 | 0.9220 | 32.41 | 0.9026 | 33.33 | 0.9389 | 39.44 | 0.9783 |
| SRPN (ours) | ×2 | **38.34** | 0.9619 | **34.29** | **0.9232** | **32.47** | **0.9032** | **33.50** | **0.9401** | **39.76** | **0.9796** |
| EDSR | ×3 | 34.65 | 0.9280 | 30.52 | 0.8462 | 29.25 | 0.8093 | 28.80 | 0.8653 | 34.17 | 0.9476 |
| RCAN | ×3 | 34.74 | 0.9299 | 30.65 | 0.8482 | 29.32 | 0.8111 | 29.09 | 0.8702 | 34.44 | 0.9499 |
| SRFBN | ×3 | 34.70 | 0.9292 | 30.51 | 0.8461 | 29.24 | 0.8084 | 28.73 | 0.8641 | 34.18 | 0.9481 |
| SAN | ×3 | 34.75 | 0.9300 | 30.59 | 0.8476 | 29.33 | 0.8112 | 28.93 | 0.8671 | 34.30 | 0.9494 |
| HAN | ×3 | 34.75 | 0.9299 | 30.67 | 0.8483 | 29.32 | 0.8110 | 29.10 | 0.8705 | 34.48 | 0.9500 |
| IGNN | ×3 | 34.72 | 0.9298 | 30.66 | 0.8484 | 29.31 | 0.8105 | 29.03 | 0.8696 | 34.39 | 0.9496 |
| CSNLN | ×3 | 34.74 | 0.9300 | 30.66 | 0.8482 | 29.33 | 0.8105 | 29.13 | 0.8712 | 34.45 | 0.9502 |
| RFANet | ×3 | 34.79 | 0.9300 | 30.67 | 0.8487 | 29.34 | 0.8115 | 29.15 | 0.8720 | 34.59 | 0.9506 |
| SRPN (ours) | ×3 | **34.84** | **0.9303** | **30.76** | **0.8497** | **29.39** | **0.8120** | **29.36** | **0.8749** | **34.88** | **0.9515** |
| EDSR | ×4 | 32.46 | 0.8968 | 28.80 | 0.7876 | 27.71 | 0.7420 | 26.64 | 0.8033 | 31.02 | 0.9148 |
| SRMDNF | ×4 | 31.96 | 0.8925 | 28.35 | 0.7787 | 27.49 | 0.7337 | 25.68 | 0.7731 | 30.09 | 0.9024 |
| RCAN | ×4 | 32.63 | 0.9002 | 28.87 | 0.7889 | 27.77 | 0.7436 | 26.82 | 0.8087 | 31.22 | 0.9173 |
| SRFBN | ×4 | 32.47 | 0.8983 | 28.81 | 0.7868 | 27.72 | 0.7409 | 26.60 | 0.8015 | 31.15 | 0.9160 |
| SAN | ×4 | 32.64 | 0.9003 | 28.92 | 0.7888 | 27.78 | 0.7436 | 26.79 | 0.8068 | 31.18 | 0.9169 |
| HAN | ×4 | 32.64 | 0.9002 | 28.90 | 0.7890 | 27.80 | 0.7442 | 26.85 | 0.8094 | 31.42 | 0.9177 |
| IGNN | ×4 | 32.57 | 0.8998 | 28.85 | 0.7891 | 27.77 | 0.7434 | 26.84 | 0.8090 | 31.28 | 0.9182 |
| CSNLN | ×4 | 32.68 | 0.9004 | 28.95 | 0.7888 | 27.80 | 0.7439 | 27.22 | 0.8168 | 31.43 | 0.9201 |
| RFANet | ×4 | 32.66 | 0.9004 | 28.88 | 0.7894 | 27.79 | 0.7442 | 26.92 | 0.8112 | 31.41 | 0.9187 |
| SRPN (ours) | ×4 | **32.72** | **0.9010** | **29.04** | **0.7917** | **27.87** | **0.7457** | **27.25** | **0.8172** | **31.82** | **0.9221** |

Table 3: Quantitative results of large SR networks. Best results are **highlighted**.

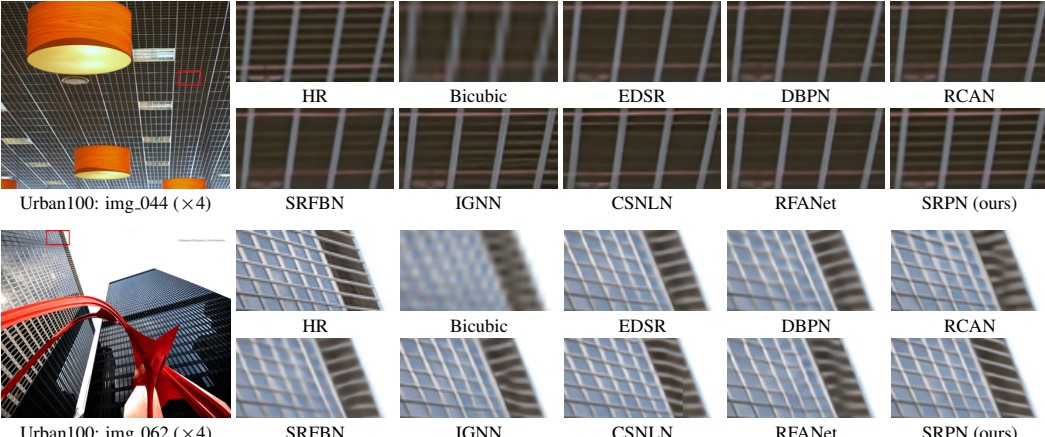

Figure 5: Visual comparison of different *large* SR networks on the Urban100 dataset (×4).

### 4.3 COMPARISONS WITH LIGHTWEIGHT NETWORKS

We compare our SRPN-Lite with lightweight SR networks: SRCNN (Dong et al., 2014), FSR-CNN (Dong et al., 2016), VDSR (Kim et al., 2016a), DRCN (Kim et al., 2016b), LapSRN (Lai et al., 2017), DRRN (Tai et al., 2017a), MemNet (Tai et al., 2017b), CARN (Ahn et al., 2018) and IMDN (Hui et al., 2019). The compared results are produced from their released code.

**Performance Comparisons.** Table 2 shows PSNR/SSIM comparisons for ×2, ×3, and ×4 SR. When compared with previous leading models, our SRPN-Lite obtains the best performance on all benchmarks and scaling factors. Unlike most comparison methods, which achieve efficiency through careful network designs, our work starts with the existing EDSR baseline (Lim et al., 2017) and prunes it to a much smaller network, showing the effectiveness of our proposed SRP.

**Model Size and Mult-Adds.** Our SRPN-Lite has the fewest parameter number in comparison to recent efficient SR works such as MemNet, CARN, and IMDN. The comparison in terms of Mult-Adds (measured when the output size is set to 3×1,280×720) is also presented. As seen, our SRPN-Lite costs fewer Mult-Adds than most comparison methods. These results demonstrate the merits of SRP against other counterparts in striking a better network performance-complexity trade-off.

| Method | Params | Mult-Adds | Set5 | Set14 | B100 | Urban100 |
|---|---|---|---|---|---|---|
| MoreMNAS-A (Chu et al., 2019b) | 1,039K | 238.6G | 37.63 | 33.23 | 31.95 | 31.24 |
| FALSR-A (Chu et al., 2019a) | 1,021K | 234.7G | 37.82 | 33.55 | 32.12 | 31.93 |
| CARN+KD (Lee et al., 2020) | 1,592K | 222.8G | 37.82 | N/A | 32.08 | N/A |
| SRPN-Lite (ours) | 609K | 139.9G | 38.10 | 33.70 | 32.25 | 32.26 |

Table 4: Model complexity comparisons ($\times 2$). Output size is $3 \times 1,280 \times 720$ to calculate Mult-Adds.

**Visual Comparisons.** The visual comparisons at $\times 4$ scale are shown in Fig. 4. It is easy to see that most of the comparison algorithms can barely recover the lost texture details in the correct direction; moreover, they suffer from blurring artifacts. In stark contrast, our SRPN-Lite effectively alleviates the blurring artifacts and recovers sharper structural fine textures in the right way, justifying the performance superiority of our SRP approach over other methods.

## 4.4 COMPARISONS WITH LARGER NETWORKS

Most previous lightweight SR networks neglect to extend their models to deeper and compare with larger networks. As stated in Sec. 3.4, we extend our SRP method to larger network training and obtain a deeper network SRPN. In Tab. 3, we compare with large SR networks: EDSR (Lim et al., 2017), DBPN (Haris et al., 2018), RCAN (Zhang et al., 2018b), SRFBN (Li et al., 2019), SAN (Dai et al., 2019), HAN (Niu et al., 2020), IGNN (Zhou et al., 2020), CSNLN (Mei et al., 2020), and RFANet (Liu et al., 2020). The compared results are produced from their released code.

**Performance Comparisons.** In Tab. 3, we observe that our SRPN achieves the best PSNR/SSIM values across different test sets and scales, except for SSIM value on Set5 ($\times 2$). Our SRPN utilizes slightly fewer parameters (15.3M vs. 15.4M) with the same network depth as RCAN (Zhang et al., 2018b). Notably, our SRPN removes the channel attention module in RCAN, but still achieves better results. This is mainly because our SRPN is initially trained from a wider and stronger network (with 96 channels). We train the much larger network and then prune it to the target size.

**Visual Comparisons.** We further show visual comparisons with larger networks (e.g., CSNLN, RFANet, and IGNN) in Fig. 5, we can also observe similar situations as in Fig. 4. For example, in img_011, other methods suffer from heavy blurring artifacts. In other cases, they may not recover clear structures in img_044 or even wrong details in img_062. Our SRPN achieves more visually pleasing results with more structural details. Those comparisons demonstrate that our SRP method can be applied to train larger networks and obtain comparable or even better performance.

## 4.5 MORE COMPARISON WITH MODEL COMPRESSION TECHNIQUES

We further compare our SRP to other representative efficient image SR approaches via model compression. Concretely, neural architecture search based methods (Chu et al., 2019b;a) and knowledge distillation (KD) based methods (Lee et al., 2020) are compared to. Quantitative results at $\times 2$ scale are presented in Tab. 4, where our SRPN-Lite delivers *better* PSNR results across different datasets with *fewer* parameters and Mult-Adds. With our SRP pruning method, there is no need to search massive network architectures or pretraining a teacher network, which usually consumes considerable computation resources. These comparisons show that our SRP, as a network *pruning* method, has as much potential (if not more) as other model compression techniques for efficient image SR.

## 5 CONCLUSION

Existing efficient image SR methods have achieved promising results with moderate model parameters and Mult-Adds. However, they neglect to extend their approaches to larger networks, which usually perform much better with more parameters and/or operations with residual learning. Meanwhile, although model compression approaches, like knowledge distillation and neural architecture search, have also been exploited for efficient image SR networks, they typically demand excessive computation resources. As another promising model compression method, network pruning is hard to be incorporated into lightweight SR models directly. To address these issues, we present structure-regularized pruning (SRP), which imposes $L_2$ regularization on the pruned structure to align the locations of pruned filters across different layers. It can transfer the expressive power in the unimportant filters to the remaining part of the network. We employ SRP to train both lightweight and large image SR networks, resulting in a lightweight network SRPN-Lite and a very deep one SRPN, delivering superior performance over recent SOTA efficient SR methods.

**Acknowledgments**. This research is supported by the U.S. Army Research Office Award W911NF-17-1-0367.

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
