# OpenReview forum: "Learning Efficient Image Super-Resolution Networks via Structure-Regularized Pruning"
_ICLR.cc/2022/Conference — ICLR 2022 Poster_

### Official Review · Reviewer_t8M1 · 2021-10-26

**Correctness:** 4
**Technical Novelty And Significance:** 3
**Empirical Novelty And Significance:** 3
**Recommendation:** 8
**Confidence:** 4

**Main Review:**

Strengths
- The problem definition is well defined, addresses the challenges and provides a possible solution
- The paper is well written, easy to follow, extensive explanations make the paper readable
- The descriptions and explanations are to the point
- Terms and concepts are well defined and explained
- Proposes a pruning method for structured pruning
- Gives two derived networks SRPN-L and SRPN and reports their results which shows quality improvement compared to other SOTA
- The SRP model can be applied to SOTA networks in a plug and play model
- Extensive experimental analysis
- Ablation study shows that the proposed method is effective at pruning and maintaining the quality

Weakness / Questions
- Were there any effort for the SR models to be trained using real-world downsampling kernels instead of bicubic?
- Why are the parameter counts not provided for the large network comparison table?
- Likewise for Mult-adds


**Summary Of The Paper:**

Although light weight SR models show promise in terms of quality while using moderate sized network, it does not extend from there into larger models, or into practical use.
Model compression techniques have also been used to reduce network size but consumes significant resources and computation.
Neural architecture search and knowledge distillation are some example techniques to use for compression but not sufficiently effective.
However, network pruning can be a viable option for effective model compression, but can be proved tricky due to its difficulty regarding pruning of the filter for residual blocks.
To mitigate the issue, the authors propose a structure regularized pruning which enforces a regularization on the pruned structure to make pruned regions aligned across different layers. For instance, the method works by selecting filters with the same indices which are connected by the same residual.
The authors also employ a L2 regularization to drive weights to zero for unimportant filters and transfer its information to other parts of the network.
This is important for minimizing performance degradation.
The authors propose a derived network via using the structure regularized pruning, namely SRPN-L and SRPN.
The proposed methods show better quality  than the latest networks quantitatively and visually.


**Summary Of The Review:**

This paper is well written, containing extensive experiments and impressive results.

---

> ### Author Response · Authors · 2021-11-21
> **Response to Reviewer t8M1 (denoted as R4)**
>
> `Q1:` Were there any effort for the SR models to be trained using real-world downsampling kernels instead of bicubic?
>
> `A1:` Yes, many efforts have also been made to image SR models with real-world downsampling kernels instead of bicubic. The readers can refer to many related works [ref3, ref4, ref5].
>
> In this work, we focus on investigating the simultaneous optimization of image SR network training and pruning. At the same time, bicubic downsampling is the most commonly-used degradation kernel. Most state-of-the-art image SR methods have reported results under the bicubic degradation setting. To show the effectiveness of our method, we choose to use the bicubic setting.
>
> Of course, different downsamplings or degradation models are also pretty important in the image SR community. It is well worth to investigate different downsamplings with network compression (e.g., network pruning) in the future work.
>
> [ref3] Zhengxiong Luo, Yan Huang, Shang Li, Liang Wang, Tieniu Tan. Unfolding the Alternating Optimization for Blind Super Resolution. NeurIPS, 2020
>
> [ref4] Longguang Wang, Yingqian Wang, Xiaoyu Dong, Qingyu Xu, Jungang Yang, Wei An, Yulan Guo. Unsupervised Degradation Representation Learning for Blind Super-Resolution. CVPR, 2021
>
> [ref5]  Jingyun Liang, Kai Zhang, Shuhang Gu, Luc Van Gool, Radu Timofte. Flow-Based Kernel Prior With Application to Blind Super-Resolution. CVPR, 2021
>
>
> `Q2:` Why are the parameter counts not provided for the large network comparison table?
> Likewise for Mult-adds
>
> `A2:` Due to limited space, we **already provided** those comparisons in the **supplementary material** Table 8. Specifically, we provide the parameter, Mult-Adds, inference time, and PSNR comparisons for the large networks in Table 8. We also show them here. When we report the FLOPs (G) (namely Mult-Adds) and inference time (s), we use input size as 3x160x160.
>
> | Large models | EDSR | RCAN | SAN | CSNLN | SRPN (ours) |
> | -----------------  | ------- |-------- | ------ | --------- | ------ |
> | Parameters (M) | 40.73 | 15.44 | 15.67 | 3.06 | 15.33 |
> | FLOPs (G) |1,042.7 | 391.9 | 400.4 | 2,245.9 | 391.8 |
> | Inference Time (s) | 0.37 | 0.85 | 1.45 | 7.14 | 0.76 |
> | PSNR (dB) on Manga109 (×2) | 39.10 | 39.44 | 39.32 | 39.37 | 39.75 |
>
> We can see our SRPN obtains similar model size, FLOPs with RCAN and SAN, but obtains better performance. When compared with CSNLN, our SRPN performs more efﬁciently.

---

### Official Review · Reviewer_xzfp · 2021-10-27

**Correctness:** 1
**Technical Novelty And Significance:** 3
**Empirical Novelty And Significance:** 2
**Recommendation:** 6
**Confidence:** 4

**Main Review:**

Strength

- The proposed method achieves better performance compared to the previous lightweight SR methods. Also, the proposed method is successfully applied to larger SR models.

- The proposed method is simple and effective.

Weakness

- The proposed method is suitable when there is a residual connection and there are few Conv layers per block. On the other hand, if there is no residual block or there are many Conv layers per block, the effect may be somewhat insignificant. There is a need to compare the proposed method with the cases of the pruned model without residual block and of the pruned model with many Conv layers in the block.

- I think the comparison with the existing Purning method is somewhat lacking. Is there any comparison with unstructured pruning other than channel pruning? Also, Is there any comparison with the pruning technique used in semantic segmentation other than SR?
--> CAP: Context-Aware Pruning for Semantic Segmentation, WACV 2021

- There is some ambiguity in the details of the proposed method. In the Pruning criterion, how many random unimportant filters are selected? In the Regularization schedule, what is the initial alpha? How pre-defined constant is decided?

- Performance improvement in Table1 is not significant. Also, the proposed method is only evaluated on only one backbone SR model. It is difficult to say that the proposed method has been generalized.

- In Table 3, there are no reports about Params and Mult-Adds.

- In Table 4, there are needs more comparisons on different datasets


**Summary Of The Paper:**

This paper proposes a Structure-Regularized Pruning (SRP) for the super-resolution task. Authors found that residual connections in SR models make it difficult to develop filter pruning methods. To resolve it, the proposed method imposes constraints on the locations of pruned filters. The constraint is to align unimportant indexes across different layers. The proposed method was achieved.


**Summary Of The Review:**

However, there are some ambiguity explanations, and the model design and hyperparameter were arbitrarily selected.

---

> ### Author Response · Authors · 2021-11-21
> **Response to Reviewer xzfp (denoted as R3) part 1**
>
> `Q1:` The proposed method is suitable when there is a residual connection and there are few Conv layers per block. On the other hand, if there is no residual block or there are many Conv layers per block, the effect may be somewhat insignificant. There is a need to compare the proposed method with the cases of the pruned model without residual block and of the pruned model with many Conv layers in the block.
>
> `A1:` Thanks for the valuable suggestions to apply our method to other cases. Following the suggestions of R3, we apply our SRP to two cases. Case 1: without residual block; Case 2: many Conv layers in the block. We report the results with different pruning ratios as follows.
>
> **Without residual block.** We use CARN-M [ref1] as an example, where there is no residual block. Please note that [ref1] also proposed CARN, which consists of several residual blocks. We provide the PSNR results on Set5 (×2) of applying SRP to CARN-M [ref1] with different pruning ratios as follows.
>
> | Pruning ratio | 0 (original) | 0.1 | 0.3 | 0.5 | 0.7 | 0.9 |
> |-|-|-|-|-|-|-|
> | CARN-M [ref1] + SRP | 37.53 | 37.50 | 37.44 | 37.37 | 37.27 | 36.84 |
>
> **Many Conv layers in the block.** We use IMDN [ref2] as an example, where there are 5 Conv layers in one residual block. We provide the PSNR results on Set5 (×2) of applying SRP to IMDN [ref2] with different pruning ratios as follows.
>
> | Pruning ratio | 0 (original) | 0.1 | 0.3 | 0.5 | 0.7 | 0.9 |
> |-|-|-|-|-|-|-|
> | IMDN [ref2] + SRP | 38.00 | 37.97 | 37.89 | 37.84 | 37.70 | 37.36 |
>
> From the above two tables, we can see that our SRP can well preserve the performance when pruning the networks with different structures from the basic residual blocks.
>
> [ref1] Namhyuk Ahn, Byungkon Kang, and Kyung-Ah Sohn. Fast, accurate, and lightweight super-
> resolution with cascading residual network. In ECCV, 2018.
>
> [ref2] Zheng Hui, Xinbo Gao, Yunchu Yang, and Xiumei Wang. Lightweight image super-resolution with
> information multi-distillation network. In ACM MM, 2019
>
> `Q2:` I think the comparison with the existing Purning method is somewhat lacking. Is there any comparison with unstructured pruning other than channel pruning? Also, Is there any comparison with the pruning technique used in semantic segmentation other than SR?
>
> -> CAP: Context-Aware Pruning for Semantic Segmentation, WACV 2021
>
> `A2:` Unstructured pruning can produce considerable sparsity (thus compression rate) but the *irregular* sparsity is very hard to turn into practical acceleration. In this paper, we target *filter pruning for acceleration* (see Page 3, Related Work: “In this paper, we focus on filter pruning for easy acceleration”; in image SR, acceleration is more *urgent* than compression given the model size is typically not big but FLOPs are quite huge). Therefore, we do not compare with unstructured pruning methods, as they are two different tracks in pruning.
> Thanks for pointing out the semantic segmentation pruning method. We are more than happy to include it for comparison, yet we may not be able to have the comparison results during this short rebuttal period. The reasons are:
>
> ⋅⋅*It is a pruning method for *semantic segmentation*, while this paper tackles image SR. They are two *different* tasks. Right now, we actually do not see how to extend this method from semantic segmentation to image SR.
>
> ⋅⋅* The CAP method depends on BN layers (here is the description of CAP in their abstract “...adaptively identify the informative channels on the cumbersome model by inducing channel-wise sparsity on the scaling factors in batch normalization (BN) layers). However, BN is known not to be very effective on image SR, thus actually not used in top-performing SR networks such as EDSR, RCAN.
>
> Given these, it is definitely *non-trivial* to apply CAP to SR (probably it will need to change the CAP method itself), so we may not be able to include the specific comparison results in our paper. Instead, given the relevance, we will cite this paper and clarify the methodology differences between our method and theirs.

---

> ### Author Response · Authors · 2021-11-21
> **Response to Reviewer xzfp (denoted as R3) part 2**
>
> `Q3:`  There is some ambiguity in the details of the proposed method. In the Pruning criterion, how many random unimportant filters are selected? In the Regularization schedule, what is the initial alpha? How pre-defined constant is decided?
>
> `A3:` Thanks for asking for those details that could be important to further improve our method. We explain them as follows.
>
> **Random unimportant filters selection.** This is mentioned in the Sec. 3.4 Implementation Details. For lightweight network, 211 out of 256 filters are selected. For very deep network, 32 out of 96 filters are selected.
>
> **Initial alpha.** The initial alpha is set to 0.
>
> **Pre-defined constant.** $\Delta$ describes the penalty increment. Since we desire a smooth and gentle regularization process, this constant cannot be too large. We empirically set it to 1e-4 referring to the normal weight decay value in classification (e.g., for ResNet50 on ImageNet, the weight decay value is 1e-4).
>
> `Q4:` Performance improvement in Table1 is not significant. Also, the proposed method is only evaluated on only one backbone SR model. It is difficult to say that the proposed method has been generalized.
>
> `A4:` We **already provided** results for generalization of SRP in **supplementary material**. We apply our SRP to some classic image SR structures, like IMDN and CARN. We provide the results on Set5 (×2) of applying SRP to IMDN and CARN in Table 9. We also show Table 9 here.
>
> | Pruning ratio | 0 (original) | 0.1 | 0.3 | 0.5 | 0.7 | 0.9 |
> |-|-|-|-|-|-|-|
> | CARN+SRP | 37.76 | 37.74 | 37.65 | 37.61 | 37.49 | 37.05 |
> | IMDN+SRP | 38.00 | 37.97 | 37.89 | 37.84 | 37.70 | 37.36 |
>
>
> `Q5:` In Table 3, there are no reports about Params and Mult-Adds.
>
> `A5:` Due to limited space, we **already provided** those comparisons in the **supplementary material**. Specifically, we provide the parameter, Mult-Adds, inference time, and PSNR comparisons for the large networks in Table 8. We also show them here. When we report the FLOPs (G) (namely Mult-Adds) and inference time (s), we use input size as 3x160x160.
>
> | Large models | EDSR | RCAN | SAN | CSNLN | SRPN (ours) |
> | -----------------  | ------- |-------- | ------ | --------- | ------ |
> | Parameters (M) | 40.73 | 15.44 | 15.67 | 3.06 | 15.33 |
> | FLOPs (G) |1,042.7 | 391.9 | 400.4 | 2,245.9 | 391.8 |
> | Inference Time (s) | 0.37 | 0.85 | 1.45 | 7.14 | 0.76 |
> | PSNR (dB) on Manga109 (×2) | 39.10 | 39.44 | 39.32 | 39.37 | 39.75 |
>
>
> `Q6:` In Table 4, there are needs more comparisons on different datasets
>
> `A6:` Thanks for the valuable suggestion. We have added results on **more datasets** (like Set14 and Urban100) for Table 4.
>
> | Method | Params | Mult-Adds | Set5 | Set14 | B100 | Urban100 |
> | -- | -- | -- | -- | -- | -- | -- |
> | MoreMNAS-A  | 1,039K | 238.6G | 37.63 | 33.23 | 31.95 | 31.24 |
> | FALSR-A | 1,021K | 234.7G | 37.82 | 33.55 | 32.12 | 31.93 |
> | CARN+KD | 1,592K | 222.8G | 37.82 | N/A | 32.08 | N/A |
> | SRPN-L (ours) | 609K | 139.9G | 38.10 | 33.70 | 32.25 | 32.26 |
>
> We can see our SRPN-L still achieves the best performance on those standard datasets. Those comparisons show the effectiveness of our method.

---

### Official Review · Reviewer_yDGk · 2021-10-29

**Correctness:** 3
**Technical Novelty And Significance:** 4
**Empirical Novelty And Significance:** 4
**Recommendation:** 8
**Confidence:** 5

**Main Review:**

Strengths:
First of all, I like this idea, which jointly optimizes the network pruning and image SR. The results are impressive and show promising potential for future efficient works.
1. Making effective SR models more efficient is of interest to a broad audience in low-level vision. This paper tackles this issue thus potentially can have a big impact.
2. In terms of methodology, they present a new network pruning method (SRP) for efficient SR based on regularization, which is technically sound.
3. Empirically, they apply their method to both large SR networks and lightweight networks. In both cases, they show their pruned networks achieve the best performance with fewer parameters or Mult-Adds. The results look pretty strong.
4. The supplementary material contains pretty much information. Most of my concerns, like differences with other related works, can be addressed by the supplementary file.
5. The writing and organization are good. The whole paper, including the supplementary, is well prepared.
6. The authors provide demo code to reproduce the results in the paper, which makes the paper more reliable and convincing.

Weaknesses:
In general, this paper is well-written and has extensive experiments with strong results. I have the following questions about the method and result details.
1. About the method, applying L2 regularization for sparsity appears not a common practice since it cannot make weights exactly zero. L1 regularization (i.e., lasso) is more normal in statistics in the sense of imposing sparsity. The discussion in “Regularization Form” seems not enough to treat this question properly. The authors are highly suggested to explain more.
2. I noted in Tab. 1, the listed “pruning ratio” seems not aligned with the parameter reduction. E.g., for pruning ratio 0.5, the compression ratio should be 2. While by the parameters, the compression ratio is 1369.9/381.8=3.6, much larger than 2. Why this?
Btw, typos: Appendix “Line 756-760 in the main paper” seems not correct.

**Summary Of The Paper:**

The authors propose structure-regularized pruning (SRP) for efficient image super-resolution. They provide a general idea to structurally prune both lightweight and large image SR networks. Extensive results are provided to support their claimed contributions.

**Summary Of The Review:**

The idea is simple yet efficient for both lightweight and large image SR networks. The ablation study, like the visualization of pruning process, demonstrates the effect of the method. The main comparison results with others are impressive.

--------------------------------
The authors addressed my concerns in the reply. After considering other reviews and response, I decide to keep my initial score and vote for acceptance.

---

> ### Author Response · Authors · 2021-11-21
> **Response to Reviewer yDGk (denoted as R2)**
>
> `Q1:` About the method, applying L2 regularization for sparsity appears not a common practice since it cannot make weights exactly zero. L1 regularization (i.e., lasso) is more normal in statistics in the sense of imposing sparsity. The discussion in “Regularization Form” seems not enough to treat this question properly. The authors are highly suggested to explain more.
>
> `A1:` Under the normal regularization strength (like 1e-3~1e-4), L1 regularization achieves more real sparsity than L2 regularization indeed. Yet, in our paper, we employ the regularization strength in a *gradually arising* fashion (see Eq. (5)). The regularization strength finally reaches a pretty large amount (0.5). Under this very strong strength, L2 regularization can achieve a similar practical role to L1 regularization in terms of sparsity.
>
> On the other hand, for a specific parameter, the gradient of L2 regularization depends on its magnitude. It can work as a *self-adaptive scaling* for different weights, which is beneficial to the training stability and makes it much easier for us to set hyper-parameters (such as the $\Delta$ in Eq. (5)). Taking all these into consideration, we choose L2 regularization over L1.
>
> Thanks to R2 for letting us know this choice is not clearly explained in our paper. We shall add the above explanation in the new version.
>
> `Q2:` I noted in Tab. 1, the listed “pruning ratio” seems not aligned with the parameter reduction. E.g., for pruning ratio 0.5, the compression ratio should be 2. While by the parameters, the compression ratio is 1369.9/381.8=3.6, much larger than 2. Why this? Btw, typos: Appendix “Line 756-760 in the main paper” seems not correct.
>
> `A2:` Thanks for noticing these interesting details and pointing out some tytos. We explain them as follows.
>
> **Compression ratio.** The pruning ratio in Tab. 1 is a *layer-wise* pruning ratio, not for the *whole network*. The pruning ratio (or sparsity) for the whole network cannot be directly inferred from the layer-wise pruning ratio. This is exactly why we also list the specific Params in Tab. 1 so that the readers can know precisely how many parameters are reduced.
>
> **Typos.** Thanks for pointing out these typos. We have corrected them and conducted proofreading several times.

---

> > ### Comment · Reviewer_yDGk · 2021-11-26
> > **After rebuttal**
> >
> > Thanks for the response, which addresses my concerns.
> >
> >  I like the simple yet efficient idea, which jointly optimize the image SR training and model compression (i.e., network pruning). I think this idea would inspire others to investigate more and deeper for efficient image SR.
> >
> > The experiments are also very extensive, especially the quantitative/visual comparisons and pruning visualizations. Many additional results (including demo code) are provided in the supplementary file and further make this work solid.
> >
> > I also read other reviewers’ comments and the authors’ corresponding responses. I would like to keep my initial score and vote for acceptance.

---

> > > ### Author Response · Authors · 2021-11-27
> > > **Thanks Reviewer yDGk for approving our work**
> > >
> > > Thanks for agreeing with the idea, experiments, and potentials of our work.
> > >
> > > We will further improve this paper by mentioning some experiments in the supplementary file (as Reviewer xzfp gives this suggestion).

---

### Official Review · Reviewer_N1G5 · 2021-11-02

**Correctness:** 3
**Technical Novelty And Significance:** 2
**Empirical Novelty And Significance:** 3
**Recommendation:** 5
**Confidence:** 4

**Details Of Ethics Concerns:**

no ethics concern

**Main Review:**

Strength:
1 the paper is well-written and easy to read.
2 the proposed method is simple yet effective, where the experimental results on both lightweight and large SR networks demonstrate state-of-the-art results.

Weakness:
1 In general, this paper lacks novelty. Filter pruning, as well as the regularization term and schedule, are common techniques in high-level vision tasks, and this paper applies them to the SR task. The difference might be the pruning criterion. Indeed, the extensive residual connections cause the main difficulty to directly apply these techniques to the SR task. However, the authors do not include deep investigations to this problem, e.g., experiments of direct application, insightful analysis about the intrinsic reasons, etc. In addition, to solve this problem, the authors randomly select a set of filters to be pruned and fix them during pruning. This may affect the stability of training, where no repetitive experiments are provided.
2 In Table 3, the SRPN is pruned from an extended version of RCAN (with 96 channels) and achieves state-of-the-art performance. This is logical since the extended RCAN is more powerful and contains redundant parameters. It is better to, the performance of the extended RCAN should also be provided.
3 The authors claim a general idea to prune SR networks, especially for large networks. However, only the pruning of RCAN (without channel attention) is validated. To demonstrate the generalization capacity, more networks with different topologies should be included.


**Summary Of The Paper:**

This paper proposes a structure-regularized filter pruning strategy for efficient SISR. Considering the residual connections in the SR networks, the authors propose to prune the filters that are aligned across layers connected by the same residual. The weight of those layers are optimized using an L-2 regularization to avoid performance drop. Experiments on both lightweight and large SR networks are conducted, with superior performance both quantitatively and qualitatively.

**Summary Of The Review:**

This paper is clear and well-written. However, the novelty of the paper is somehow limited. Also, there are limited insightful investigations about the idea and intrinsic motivations. Finally, some experiments are missed.

---

> ### Author Response · Authors · 2021-11-19
> **Response to Reviewer N1G5 (denoted as R1) part 1**
>
> `Q1:` In general, this paper lacks novelty. Filter pruning, as well as the regularization term and schedule, are common techniques in high-level vision tasks, and this paper applies them to the SR task. The difference might be the pruning criterion. Indeed, the extensive residual connections cause the main difficulty to directly apply these techniques to the SR task. However, the authors do not include deep investigations to this problem, e.g., experiments of direct application, insightful analysis about the intrinsic reasons, etc. In addition, to solve this problem, the authors randomly select a set of filters to be pruned and fix them during pruning. This may affect the stability of training, where no repetitive experiments are provided.
>
> `A1:` **For the novelty.** We cannot agree with R1 about the lack of novelty argument. The main technical problem for applying pruning methods in classification to SR lies in the extensive SR blocks. Although _regularization and filter pruning_ are widely used in classification, they are generic concepts.  How to materialize these generic concepts into a concrete effective algorithm in SR is _non-trivial and has not been effectively done by any previous paper_, while our paper bridges this gap.
>
> **For the “experiments of direct application”.**  We **already show** directly applying the (most representative) filter pruning method  in classification (i.e., the L1-norm pruning (Li et al., 2017) )  to SR is not as effective. The analysis is also provided in our ablation study (Tab. 3, Fig. 1).
>
> Besides, the **intrinsic reason** that directly applying L1 pruning to SR does not work well is also discussed in the ablation study -- see page 7 “However, our results are significantly better than theirs. The reason is that...”. We hope R1 can notice these results and analyses and have a more fair evaluation.
>
> R1 might misunderstand this paper. The pruning criterion used in this paper is simply L1-norm (see Page 7 “Notably, our method also adopts the L1-norm as pruning criterion, the same as (Li et al., 2017)”), which is actually the most common and widely-used pruning criterion (this is also the reason we did not claim any contribution from the pruning criterion side). Thus, R1’s comment** “The difference might be the pruning criterion” **is a misunderstanding.
>
> **For the random selection** of pruned filters for constrained conv layers, **(1)** methodologically, because we employ regularization instead of one-shot pruning (e.g., the L1-norm pruning), the training process is not unstable owing to the smooth nature of regularization. **(2)** Empirically, in our experiments, we do not observe any unstable training, either.
>
> As for the comment **“no repetitive experiments are provided”**, R1 might overlook the results in the Appendix Tab. 11, where we **already provided** the multi-run results with different random seeds. The small standard deviations in Tab. 11 also imply that there is no instability in our training.
>
> Random selection of pruned filters may appear problematic at first sight, while it works pretty well practically, thanks to our regularization design. In our view, this is actually a proof that shows the effectiveness of our method.
>
> We still show the performance and statistical variation with different random seeds.
>
> |Pruning ratio | 0.1 | 0.3 |  0.5 | 0.7 | 0.9 |
> | - | - | - | -| - | - |
> | L 1 -norm | 37.90 (0.01) | 37.80 (0.02) | 37.75 (0.03) | 37.59 (0.02) | 36.88 (0.02) |
> | SRPN (ours) | 37.98 (0.02) | 37.89 (0.01) | 37.86 (0.02) | 37.70 (0.01) | 37.30 (0.01) |

---

> ### Author Response · Authors · 2021-11-19
> **Response to Reviewer N1G5 (denoted as R1) part 2**
>
> `Q2:` In Table 3, the SRPN is pruned from an extended version of RCAN (with 96 channels) and achieves state-of-the-art performance. This is logical since the extended RCAN is more powerful and contains redundant parameters. It is better to, the performance of the extended RCAN should also be provided.
>
> `A2:`  Thanks for the valuable suggestion. We continue to train the extended RCAN with 96 channels. Let’s denote it as RCAN-F96, which consumes much more GPU memories and running time. Here, we only train RCAN-F96 with scale 2 and test it on standard datasets. We compare RCAN-F96 and our pruned version SRPN and report the PSNR (dB) in the following table.
>
> | Scale=2 | Params | Set5 | Set14 | B100 | Urban100 | Manga109 |
> | - | - | - | - | - | - | - |
> | RCAN-F96   | 34.5 M | 37.38 | 34.50 | 32.49 | 33.64 | 39.91 |
> | SRPN (ours)| 15.3 M | 38.34 | 34.29 | 32.47 | 33.50 | 39.76 |
>
> We can see from the above table that RCAN-F96 performs better overall. But, it has about 34.5 M parameters, being much larger than our SRPN’s model size 15.3M.
>
> `Q3:` The authors claim a general idea to prune SR networks, especially for large networks. However, only the pruning of RCAN (without channel attention) is validated. To demonstrate the generalization capacity, more networks with different topologies should be included.
>
> `A3:`  We **discussed** the generalization ability of our SRP method and **already provided** results for generalization of SRP in Appendix Tab. 9. We apply our SRP to some classic image SR structures, like IMDN and CARN. We provide the results on Set5 (×2) of applying SRP to IMDN and CARN in Table 9. We also show Table 9 here.
>
> | Pruning ratio | 0 (original) | 0.1 | 0.3 | 0.5 | 0.7 | 0.9 |
> |-|-|-|-|-|-|-|
> | CARN+SRP | 37.76 | 37.74 | 37.65 | 37.61 | 37.49 | 37.05 |
> | IMDN+SRP | 38.00 | 37.97 | 37.89 | 37.84 | 37.70 | 37.36 |
>
> IMDN and CARN have different topologies from the basic EDSR baseline. We can see our SRP method still performs well in those structures.

---

> ### Author Response · Authors · 2021-11-27
> **Further discussions with Reviewer N1G5 (denoted as R1)**
>
> Dear Reviewer N1G5,
>
> We thank you for the precious review time and valuable comments. We have provided corresponding responses and results, which we believe have covered your concerns.
>
> We hope to further discuss with you whether your concerns have been addressed or not. If you still have any unclear parts of our work, please let us know. Thanks.
>
> Best,
>
> Authors

---

> > ### Comment · Reviewer_N1G5 · 2021-11-29
> > **After rebuttal comments**
> >
> > Thanks for your response which solves most of my concerns. While this paper is acknowledged as simple yet empirically effective, I still recommend the authors include more insightful investigations/analyses for further improvements. Pruning/accelerating the low-level vision tasks is of great significance in practical applications, and this paper may inspire some future works in the community.

---

> > > ### Author Response · Authors · 2021-11-29
> > > **Thanks Reviewer N1G5 for approving our work**
> > >
> > > Dear Reviewer N1G5,
> > >
> > > Thanks for agreeing that our response solves most of the concerns and our paper may inspire some future works.
> > >
> > > Currently, we have provided extensive results and analyses in the main paper and supplementary file.
> > >
> > > In our future work, we would like to include more insightful investigations and analyses.
> > >
> > > Best,
> > >
> > > Authors

---

### Author Response · Authors · 2021-11-29
**Response to all reviewers and area chairs**

We thank all reviewers and area chairs for their valuable time and comments. After discussing with reviewers and providing more clarifications/results/analyses, we would like to give a brief response.

All reviewers now agree with the novelty, extensive experiments (e.g., ablation study, main comparisons, and supplementary file), and writing/organization of our paper.

Reviewer yDGk (denoted as R2), Reviewer xzfp (denoted as R3), and Reviewer t8M1 (denoted as R4) all hold a **positive** side for our work. Our responses have covered their questions.

Although Reviewer N1G5 (denoted as R1) gives a negative score in the first round, R1 agrees that we have solved most of her/his concerns and our method would inspire related future works. R1 mainly suggests that we should include more insightful investigations/analyses for further improvements. We promise to further improve our work in the future.

We have submitted the demo code of this paper. We would make all the code, trained models, and results available to the public soon. We hope to further investigate this direction in low-level vision together with other researchers.

We thank all reviewers and area chairs again!

Best,

Authors

---

### Decision · Program_Chairs · 2022-01-20

**Decision:**

Accept (Poster)

**Comment:**

This paper receives positive reviews. The authors provide additional results and justifications during the rebuttal phase. All reviewers find this paper interesting and the contributions are sufficient for this conference. The area chair agrees with the reviewers and recommends it be accepted for presentation.